# Refined weathering CO$_2$ budget of the Tibetan Plateau strongly modulated by sulphide oxidation

Wenjing Liu[1,2], Zhifang Xu [1,2] ✉, Huiguo Sun[1,2], Mingyu Zhao [1,2], Yifu Xu [1,2] & Zhengtang Guo [1,2]

Estimation of net CO$_2$ consumption by weathering in orogen is complicated as high erosion rate promotes competing processes of CO$_2$ consumption (silicate weathering) and releasing (sulfuric acid (H$_2$SO$_4$) dissolution of carbonate). Quantification of H$_2$SO$_4$ disturbing on weathering is missing in the Tibetan Plateau, hindering the understanding of Himalayan orogenesis impact on global carbon cycle. Here we calculate the riverine solute contributions from both carbonic and sulfuric acid mediated weathering, and their weathering fluxes with major river geochemistry dataset from the Tibetan Plateau. We find that silicate weathering is not anomalous, while carbonate weathering flux is 2.09% of the global value with 1.01% drainage area. Over 80% H$_2$SO$_4$ originated from pyrite oxidation is consumed by carbonate weathering, which counteracts ~58% of the CO$_2$ consumption flux by silicate weathering. The refined weathering CO$_2$ budget in this work provides quantitative modern evidence for pyrite weathering in orogen serving as negative feedback on atmospheric pCO$_2$.

The formation of the Tibetan Plateau exerts profound impacts on Earth's surface environment evolution in the Cenozoic. Particularly, it has been a hotspot for continental weathering studies since the proposing of "uplift-weathering hypothesis"[1,2], which ascribes the atmospheric pCO$_2$ drawdown and global cooling during the Cenozoic to the enhanced silicate weathering (SW) and CO$_2$ consumption in the Himalayan-Tibetan orogeny. However, later studies questioned it[3–6] and proposed that the impact of plateau formation on continental weathering and atmospheric pCO$_2$ level was limited[7–9]. Key controversies lie in the negative feedback mechanism of the Uplifting-Weathering Hypothesis. Sulfuric acid (H$_2$SO$_4$) generated in oxidative weathering of pyrite (OWP) serves as a proton provider in carbonate weathering, which has been argued as an important CO$_2$ source counteracting CO$_2$ consumption by SW at different scales recently[10–16]. The high erosion rate on the Tibetan Plateau resulted from uplifting, and the largest distribution area of low-altitude glaciers makes it a hotspot for H$_2$SO$_4$ generation by OWP, and thus a CO$_2$ source counteracting CO$_2$ consumption by silicate weathering, which might serve as an important negative feedback mechanism of atmospheric pCO$_2$ in Cenozoic. However, quantitative and systematic evidence, especially for the relative importance of different weathering pathways and the functions of different acid agents, is missing.

Hydro-geochemistry of river water is informative of weathering process, fluxes, and CO$_2$ budgets as it integrates the weathering products over the whole drainage area of the river networks. Regional case studies on basin weathering have been conducted for the Tibetan Plateau, revealing that the solute sources and weathering fluxes of Himalayan–Tibetan river basins vary considerably through the plateau[17–22]. The river solute source identification and quantification for large rivers in the Tibetan Plateau are not well constrained as significant spatial variations exist on both weathering bedrock and weathering acid agents, and systematic datasets to calculate the riverine ion contribution from different weathering reactions are lacking. Investigation of the fluvial geochemistry is needed to gain a quantitative estimation of the chemical weathering and associated CO$_2$ fluxes for the plateau as a whole. A knowledge gap exists when evaluating the

[1]State Key Laboratory of Lithospheric and Environmental Coevolution, Institute of Geology and Geophysics, Chinese Academy of Sciences, Beijing, China. [2]College of Earth and Planetary Sciences, University of Chinese Academy of Sciences, Beijing, China. ✉e-mail: zfxu@mail.iggcas.ac.cn

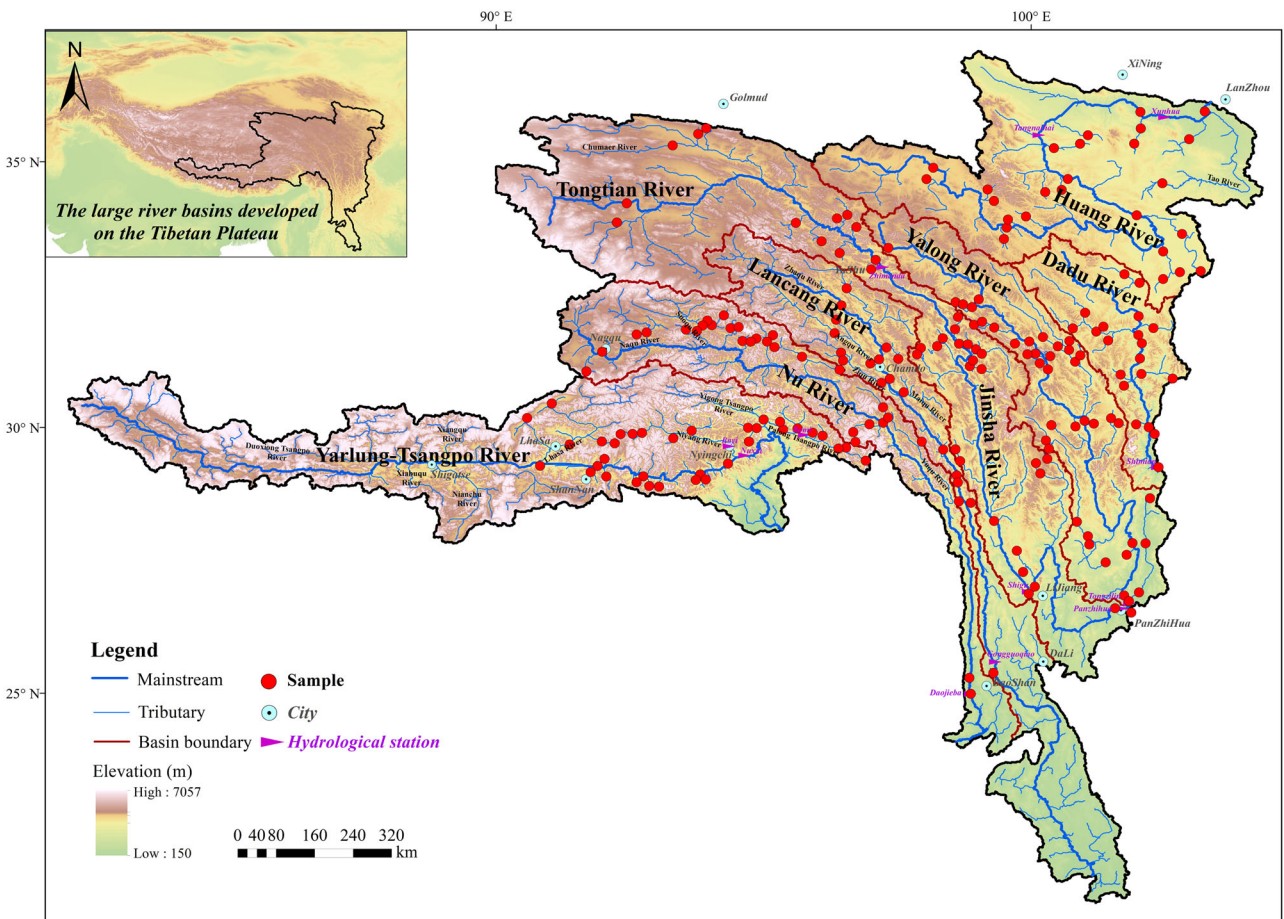

**Fig. 1 | Major river basins on the Tibetan Plateau and the sampling locations in this study.** The panel on the upper left corner shows the location of the studied area, and the main panel presents the major river basins on the Tibetan Plateau and the sampling sites. The blue lines represent the river channels, and the red lines represent the basin boundaries. Solid red dots show the sampling locations and purple triangles noted the hydrological stations. Light blue dots with black center noted the major cities in the studied area.

effect of $H_2SO_4$ on chemical weathering as sulfate contributions are complicated from atmosphere deposition, evaporite, hot spring, and OWP in the plateau[23]. In particular, OWP-sourced $H_2SO_4$, which is a strong competition acid agent with carbonic acid in chemical weathering reactions, has not been quantitively estimated yet, thus obstructing the estimation of its disturbance on the $CO_2$ budget of plateau weathering.

We systematically sampled all the major river networks on the Tibetan Plateau (Fig. 1, detailed in Supplementary Text A), including the main channels and tributaries of Nu Jiang River (upper reaches of Salween River), Yarlung Tsangpo River (upper reaches of Brahmaputra River), the upper Yellow River, the upper Yangtze River and Lantsang Jiang River (upper reaches of Mekong River) in the high flow period. Riverine hydrochemistry and multiple isotope systems ($^{87}Sr/^{86}Sr$, $\delta^{13}C_{DIC}$, $\delta^{34}S_{SO4}$, and $\delta^{18}O_{SO4}$) have been analyzed to partition riverine solute sources and to estimate the carbonic and sulfuric acid agent supply in chemical weathering reactions of different rock types and their $CO_2$ effects. This work quantifies the net weathering $CO_2$ sequestration fluxes considering the roles of both carbonic and sulfuric acid and reveals the strong modulating effect of the non-carbonic acid on weathering $CO_2$ budget of the Tibetan Plateau.

## Results and discussion
### Solute geochemistry and weathering source quantification of the Tibetan Plateau rivers
The field parameter, ion content, and isotopic compositions of river waters are tabulated in Table S1. Total dissolved solid (TDS) of large

rivers in the Tibetan Plateau averages at 215.8 mg L$^{-1}$, doubled of the global mean (100 mg L$^{-1}$, ref. 24). The ionic composition is dominated by $Ca^{2+}$, $Mg^{2+}$, and $HCO_3^-$ (Fig. S1), and the concentrations of which are distinctively higher than the rivers on the southern slope of Himalayas, such as Indus River, Ganges and Brahmaputra river system[25,26]. The frequency distributions of Sr concentrations and $^{87}Sr/^{86}Sr$ ratios for the studied rivers and surrounding area are presented in Fig. S2. The plateau rivers are with significantly higher Sr concentration and similar $^{87}Sr/^{86}Sr$ range (averaging at 3.13 μmol L$^{-1}$ and 0.7111 for the mainstream samples) compared with the global mean (0.89 μmol L$^{-1}$ and 0.71253, refs. 24,27), slightly higher than the value range of carbonate (0.708–0.710, refs. 24,28), but significantly lower than the well documented normal silicate endmember value range (0.72–0.73, refs. 28–30). The plots of $^{87}Sr/^{86}Sr$ and cation ratio index in Fig. S3 showed that most samples are distributed around the carbonate dissolution end-members, suggesting a highlighted carbonate-dominated weathering regime on the plateau.

The rivers are characterized with both higher $\delta^{13}C_{DIC}$ and lower $\delta^{34}S_{SO4}$ values (averaging at −8.4‰ and 2.6‰, respectively) than the global average (−11.6‰ and 4.4‰; refs. 13,31,32). Meanwhile, the headwaters of the Yellow, Jinsha, Lancang and Nu rivers in the arid interior region of the plateau, are with more positive $\delta^{13}C_{DIC}$ and $\delta^{34}S_{SO4}$ values, accompanying with higher $Na^+$, $Cl^-$, $Ca^{2+}$ and $SO_4^{2-}$ concentration, which is the typical feature of river water chemistry resulted from halite and gypsum dissolution. The $\delta^{34}S_{SO4}$ shows a generally decreasing trend down the mainstream of the studied rivers (Table S1), suggesting an increasing portion of OWP originated $SO_4^{2-}$

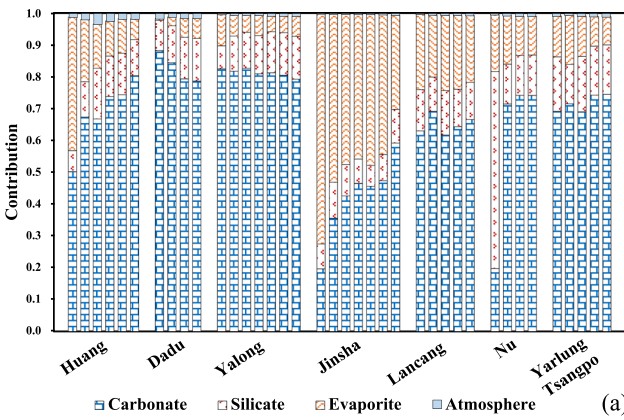
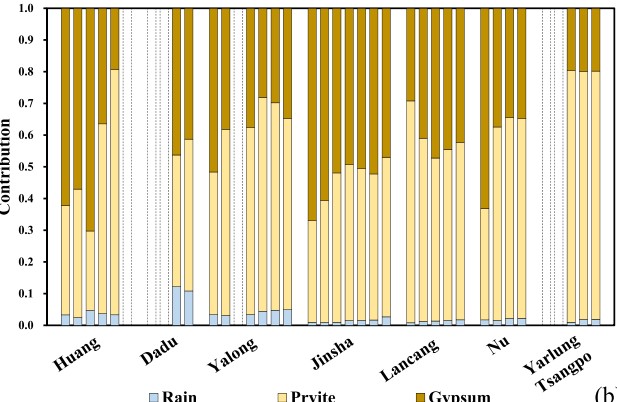

**Fig. 2 | Calculated contributions from different reservoirs to the riverine ion-loads. a** The total cationic loads (the sum of $Na^+$, $K^+$, $Ca^{2+}$, and $Mg^{2+}$); **b** riverine sulfate ($SO_4^{2-}$) for the main channels of the large rivers draining in the Tibetan Plateau. The bars represent sampling locations downward the main channels from left to right within a river.

with a more negative $\delta^{34}S_{SO4}$ (endmember value discussion detailed in the Supporting Text B III section). Among the plateau large rivers, the Yarlung-Tsangpo River flowing in the south has the highest $\delta^{13}C_{DIC}$ and lowest $\delta^{34}S_{SO4}$ mean value at −7.1‰ and +0.11‰, respectively, which implicates potential enhancement of OWP originated $H_2SO_4$ participation in weathering.

Quantifying the contributions of different riverine solute sources is pre-requisite before deriving chemical weathering rates and associated $CO_2$ consumption at the basin scale[24,26,33–36]. A forward model based on mass budget equations of cations from different sources is conducted to quantify the solute contribution (calculation procedures and results detailed in Supplementary Text B). We thoroughly consider the end members in the plateau (atmospheric precipitation, anthropogenic input, carbonate and silicate rock weathering, evaporite dissolution, and sulfide mineral weathering). The results indicate that carbonate weathering (CW) and evaporite dissolution contributions to the riverine solutes are highly variable, with an average portion at 72.8% (ranging from 6.5 to 97.2%) and 8.9% (ranging from 0 to 92.9%) of the total cationic loads in the river systems on the plateau (Table S3). In comparison, the contribution from SW is much lower and less variable. Specifically, 93% of the sampling site has SW contribution of the total cationic loads lower than 30%, with an averaging value at 17.0% (Table S3). Calculated contributions for riverine cations and sulfate from different reservoirs for the main channels are presented in Fig. 2. A new end member of sulfide mineral weathering has been brought to evaluate the sulfuric acid involvement in weathering, which has not been systematically quantified for the plateau as a whole previously. Considering both the highlighted evaporites and OWP input of riverine $SO_4^{2-}$ in the plateau, it is critical to distinguish their contribution respectively when estimating basin weathering flux and carbon effects, as evaporites dissolution is carbon neutral while OWP generates $H_2SO_4$, which dissolves carbonate and releases $CO_2$. Riverine sulfate origins are calculated with $SO_4^{2-}$ concentration and $\delta^{34}S_{SO4}$ value (Table S3, calculation detailed in the Supporting Text B III). We estimate that about 25.0 to 92.1% (averaging at 56.0%) riverine sulfate originate from OWP, which is 11% higher than the estimation for global river (45%, ref. 12). The higher $f_{pyrite}$ (the fraction of riverine sulfate originated from OWP) values for the studied river systems indicate significant sulfuric acid involvements in the plateau weathering.

## Chemical weathering rates and fluxes of the plateau
The chemical weathering fluxes and rates for individual river basins are calculated with the results of solute source quantification above, the basin area, and the discharge of different hydrology stations on the main channel of individual large rivers studied (Table 1 and

Fig. 3)[12,24–26,28,37–44]. The area-averaged chemical weathering rate of the Tibetan Plateau is calculated at 41.7 t km$^{-2}$ a$^{-1}$. As the upper reaches of the major rivers draining on the Asian continent, the large river basins in the plateau doubled the average total weathering rates of the global values (21 t km$^{-2}$ a$^{-1}$, estimated by ref. 45, and 24 t km$^{-2}$ a$^{-1}$ estimated by ref. 24). However, the enhanced weathering rate in the plateau is discriminated for CW and SW.

The SW flux of the studied river basins is 0.71% of the global value, and their area percentage is 1% of the global river basin area. The two values are 10.85% and 6.6%, respectively, when the downstream basin area is included (Table 1). The large spatial comparison indicates that the plateau river basins have significantly lower SWR (silicate weathering rate), which is only 21–55% of their lower reaches (Upper Huang, Jinsha, Nu, Lancang and Yarlung Tsangpo River, Table 1 and Fig. 3). Therefore, our calculations show that the Tibetan Plateau does not have an obvious enhancement of SWR as previous thought. Instead, the silicate materials on the plateau probably only experience initial chemical alterations with high erosion rates, relative short water and material residential time there, while longer materials residential time together with the monsoonal climate with higher mean annual temperature (MAT) and precipitation (MAP) downstream would facilitate the chemical weathering of silicate materials in the downward floodplain. We speculate that the uplifting of the plateau acts as a factory of fresh silicate materials by high erosion rates, which constantly supply sediments to the floodplain locating in the prevailing area of Asian monsoon and the enhancement of SW happens in the downstream plain area of large river basins originated from the plateau, instead of on the plateau in-situ.

Different from SW, the calculation shows that the plateau has disproportionately higher CW fluxes (2.09% of the global CW fluxes with 1.0% drainage area, Table 1). The CWR (carbonate weathering rate) of river basins on the plateau are comparable with or even higher than the value of their downstream for Huang, Changjiang and Mekong River (Table 1 and Fig. 3). It is worthwhile to be noted that the Palong Tsangpo river basin (a major tributaries of the Brahmaputra River at the upper reach) with MAT and MAP at 8.6 °C and 871 mm, respectively,[46,47] on the eastern syntaxis of the Himalaya has the highest CWR which is comparable with the downstream Brahmaputra River at the Indian plain with much higher MAT and MAP at 17.3 °C and 1543 mm (ref. 48), respectively (47.2 ton km$^{-2}$ a$^{-1}$ vs. 47 ton km$^{-2}$ a$^{-1}$). We propose that the direct impact of high physical erosion rates on the plateau drives a more significant enhancement on CWR than SWR, while the plateau river basins cannot be considered anomalous in terms of SW in comparison to other major rivers globally. Carbonate and sulfide minerals dissolve up to three orders of magnitude faster

**Table 1 | Chemical weathering fluxes and rates of different rocks and associated CO₂ consumption for large rivers in the Tibetan Plateau**

| River | Hydrological station | Basin Area $10^3$ km² | Discharge $10^9$ m³ a⁻¹ | Flux of silicate, carbonate, total rock weathering, and OWP | | | | | Weathering rates of silicate, carbonate, total rock and pyrite | | | | | CO₂ budgets by silicate and carbonate weathering | | | | | References |
|---|---|---|---|---|---|---|---|---|---|---|---|---|---|---|---|---|---|---|---|
| | | | | Cation$_{sil}$ $10^6$ ton a⁻¹ | TDS$_{sil}$ $10^9$ ton a⁻¹ | TDS$_{carb}$ | TDS$_{total}$ | $F_{SO4}^{pyrite}$ $10^9$ mol a⁻¹ | Cation$_{sil}$ ton km⁻² a⁻¹ | SWR | CWR | TWR | SOR $10^3$ mol km⁻² a⁻¹ | CO2$_{sil}$ $10^9$ mol a⁻¹ | CO2$_{carb}$ | CO2$_{carb-H2SO4}$ | Net-CO2$_{con}$ | ΦCO2$_{net}$ $10^3$ mol km⁻² a⁻¹ | |
| Upper Huang | Tangnaihai | 122 | 20.00 | 0.18 | 0.29 | 2.41 | 3.18 | 3.39 | 1.48 | 2.38 | 19.75 | 26.06 | 27.75 | 7.65 | 22.21 | 2.89 | 4.76 | 39.0 | This study |
| | Xunhua | 145.3 | 21.42 | 0.24 | 0.41 | 3.94 | 4.54 | 5.16 | 1.64 | 2.82 | 27.14 | 31.26 | 35.51 | 10.03 | 36.75 | 4.54 | 5.49 | 37.8 | This study |
| Dadu | Shimian | 66 | 37.84 | 0.22 | 0.45 | 2.73 | 3.49 | 2.41 | 3.40 | 6.77 | 41.40 | 52.87 | 36.51 | 9.00 | 26.42 | 2.06 | 6.94 | 105.1 | This study |
| Yalong | Tongzilin | 128.4 | 60.85 | 0.37 | 0.78 | 4.74 | 6.03 | 5.17 | 2.90 | 6.07 | 36.91 | 47.00 | 40.24 | 15.54 | 45.00 | 4.41 | 11.13 | 86.7 | This study |
| Jinsha | Zhimenda | 137.7 | 12.80 | 0.25 | 0.32 | 1.75 | 5.45 | 4.00 | 1.82 | 2.35 | 12.70 | 39.56 | 29.07 | 9.71 | 15.69 | 3.06 | 6.66 | 48.3 | This study |
| | Shigu | 214.2 | 42.57 | 0.29 | 0.55 | 4.36 | 10.72 | 9.70 | 1.35 | 2.57 | 20.34 | 50.06 | 45.27 | 10.26 | 37.67 | 8.54 | 1.72 | 8.0 | This study |
| | Panzhihua | 259.2 | 56.84 | 0.44 | 0.78 | 5.51 | 12.85 | 11.18 | 1.68 | 3.01 | 21.25 | 49.57 | 43.13 | 16.16 | 48.59 | 9.58 | 6.57 | 25.4 | This study |
| Lancang | Gongguoqiao | 97.2 | 31.10 | 0.26 | 0.43 | 3.18 | 4.98 | 9.07 | 2.68 | 4.39 | 32.74 | 51.24 | 93.27 | 9.31 | 25.65 | 7.67 | 1.63 | 16.8 | This study |
| Nu | Daojie | 110.2 | 53.10 | 0.31 | 0.55 | 3.83 | 5.28 | 11.32 | 2.83 | 5.02 | 34.78 | 47.92 | 102.75 | 10.89 | 31.00 | 9.63 | 1.26 | 11.4 | This study |
| Palong Tsangpo | Bomi | 9.55 | 10.64 | 0.02 | 0.04 | 0.45 | 0.58 | 0.70 | 1.88 | 4.33 | 47.20 | 60.98 | 73.47 | 0.58 | 3.97 | 0.65 | −0.07 | −7.4 | This study |
| Yarlung Tsangpo | Nuxia | 173 | 45.42 | 0.24 | 0.46 | 2.62 | 3.51 | 9.25 | 1.37 | 2.68 | 15.15 | 20.31 | 53.46 | 7.92 | 19.14 | 7.66 | 0.26 | 1.5 | This study |
| LRB-TP | | 988.9 | 317.2 | 2.10 | 3.90 | 27.0 | 41.3 | 54.3 | 2.12 | 3.95 | 27.3 | 41.7 | 54.9 | 79.4 | 236.5 | 46.2 | 33.2 | 33.6 | This study |
| Yellow R. | | 752 | 28.3 | 1.52 | 7.46 | 18.4 | 27.4 | | 2.4 | 9.9 | 24.5 | 36.4 | | 26.2 | 100.5 | | | | Refs. 37,38 |
| Changjiang | | 1705 | 899 | 4.10 | 14.6 | 62.0 | 73.0 | | | 8.56 | 14 | 22.6 | | 191 | 646 | | | | Ref. 28 |
| Salween | | 325 | 211 | 1.13 | 2.99 | 23.9 | 27.0 | | 3.49 | 9.19 | 73.5 | 82.7 | | 50.2 | 248.6 | | | | Refs. 39–41 |
| Irrawaddy | | 413 | 486 | 2.73 | 7.96 | 21.0 | 29.4 | | 6.65 | 19.4 | 51.2 | 70.6 | | 104 | 132 | | | | |
| Mekong | | 795 | 467 | 3.46 | 8.11 | 21.9 | 30.0 | | 4.35 | 10.2 | 27.5 | 37.7 | | 151.8 | 227.4 | | | | |
| Ganges | | 1060 | 459 | 4.22 | 7.76 | 30.4 | 41.0 | | 3.98 | 7.31 | 28.7 | 36 | | 181.2 | 312.9 | | | | Ref. 26; Refs. 42,43 |
| Brahmaputra | | 583 | 612 | 2.60 | 7.28 | 27.4 | 36.4 | | 4.46 | 12.5 | 47 | 59.5 | | 108.4 | 283.3 | | | | Refs. 42,43 |
| Indus | | 916 | 90 | 1.60 | 3.50 | 6.6 | 16.0 | | 1.75 | 3.82 | 7.21 | 17.5 | | 54 | 59 | | | | Refs. 24,25 |
| DS-LRB-TP | | 6549 | 3252 | 21.4 | 59.7 | 212 | 280 | | | | | | | 866.8 | 2009.7 | | | | |
| Goobal | | 99,259 | 37,358 | 200 | 550 | 1290 | 2131 | 1300 | | | | | 1300 | 8700 | 12,300 | | | | Refs. 12,24 |
| LRB-TP (% of global) | | 1.00 | 0.85 | 1.05 | 0.71 | 2.09 | 1.94 | 4.17 | | | | | | 0.91 | 1.92 | | | | |
| DS-LRB-TP (% of global) | | 6.60 | 8.71 | 10.7 | 10.8 | 16.4 | 13.2 | | | | | | | 9.96 | 16.34 | | | | |

*LRB-TP* large river basins on the Tibetan Plateau, *DS-LRB-TP* downstream of large river basins originating from the Tibetan Plateau, *OWP* oxidative weathering of pyrite, *TDS* total dissolved solids, *SOR* sulfide oxidation rate, *SWR* silicate weathering rate, *CWR* carbonate weathering rate, *TWR* total weathering rate.

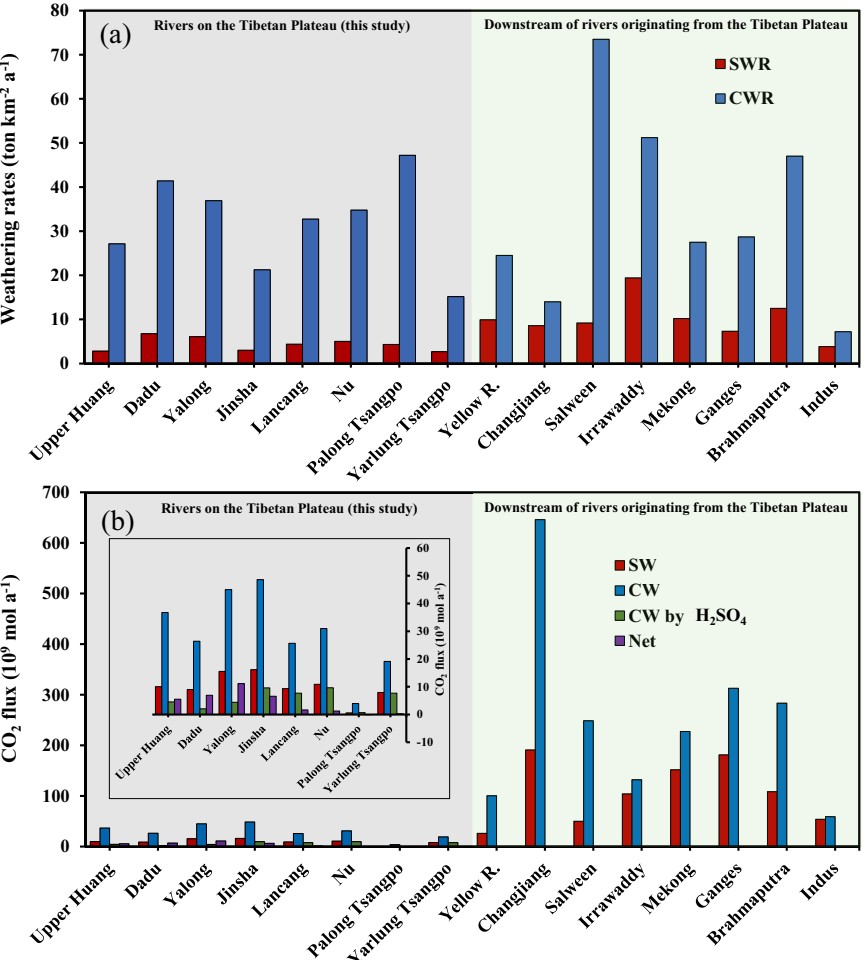

**Fig. 3 | Bar charts of weathering rates and CO₂ fluxes of large river basins originating from the Tibetan Plateau.** **a** The silicate weathering rates (SWR, red bar) and carbonate weathering rates (CWR, blue bar). **b** The CO₂ fluxes of silicate weathering (SW, red bar), carbonate weathering (CW, blue bar), carbonate weathering by sulfuric acid (CW by H₂SO₄, green bar), and the net value (purple bar).

Results and raw data for river downstream: refs. 37,38 for the downstream Huang River; ref. 28 for the Yangtze River; refs. 39–41 for the Mekong River, Salween River, and Irrawaddy River; ref. 26 and refs. 42,43 for the Brahmaputra River and Ganges River; ref. 44 for the Red River; refs. 24,25 for Indus river, respectively.

than silicate minerals[13]. The high erosion rate promotes the fresh surface generation of both carbonate and silicate minerals on the plateau basins, but the much more intensely enhanced CWR compared to SWR on the plateau is assumed to be the result of interplays between their different reacting rate with weathering fluids. Considering the different CO₂ effects of carbonate and silicate weathering with sulfuric acid, it is obligated to further evaluate the CO₂ budget of these weathering processes.

## Sulfuric acid participation and its effects on weathering CO₂ consumption budgets of the Tibetan Plateau

The Tibetan Plateau is characterized by both the enhanced OWP exposure and sulfuric acid generation due to high erosion rates resulting from tectonic and glacier activity[16,49,50], and a disproportionately strengthening of carbonate weathering revealed by river water geochemistry as discussed above. The plateau riverine DIC is dominantly contributed by carbonic and sulfuric acid weathering of carbonate as illustrated in the $\delta^{13}C_{DIC}$ and typical ionic ratios (Fig. 4 and Supplementary Text B)[16,26,51–55], indicating that OWP-originated sulfuric acid has significantly regulated the riverine DIC source and generating pathway. Strong heterogeneity of sulfuric acid disturbing on chemical weathering and CO₂ budget has been observed in previous studies at different catchment scale in the Tibetan Plateau[14,16,22]. Partitioning the riverine sulfate sources is the premise, but a knowledge gap currently

exists in the plateau to estimate the portion of protons originating from OWP and to discriminate cations released by carbonic and sulfuric acid weathering, which obstructs the evaluation of net CO₂ consumption fluxes for the plateau weathering as a whole. The flux of sulfate derived from OWP of the river networks on the Tibetan Plateau is calculated at $54.3 \times 10^9$ mol a⁻¹ (Table 1), accounting for 4.17% of the global flux of riverine sulfate sourced from OWP ($1300 \times 10^9$ mol a⁻¹, ref. 12) with a total drainage area of 1.0%. As a region with intense tectonic activity and the widest distribution of low-latitude glaciers, the Tibetan Plateau serves as one of the hotspots for pyrite exposure, which leads to the disproportionately high OWP-originated sulfate flux in the rivers there. Furthermore, we employed the simultaneous equations of $\delta^{34}S_{SO4}$ and solutes contribution from different weathering pathways to quantify sulfuric acid consumed by carbonate and silicate weathering, respectively (Supplementary Text B). The calculations show that over 80% of sulfuric acid from OWP has been consumed by carbonate weathering, and the same amount of CO₂ would be released to the atmosphere.

The long-term net CO₂ consumption budget for the major river basins on the plateau could be calculated by deducting the CO₂-releasing fluxes of H₂SO₄-carbonate reaction from CO₂ consumption fluxes of SW (detailed in the method section). The net CO₂ consumption rate ($\Phi CO_{2net}$) is the highest at $105.1 \times 10^3$ mol km⁻² a⁻¹ in the Dadu river basin at the furthest east edge of the plateau and decreases

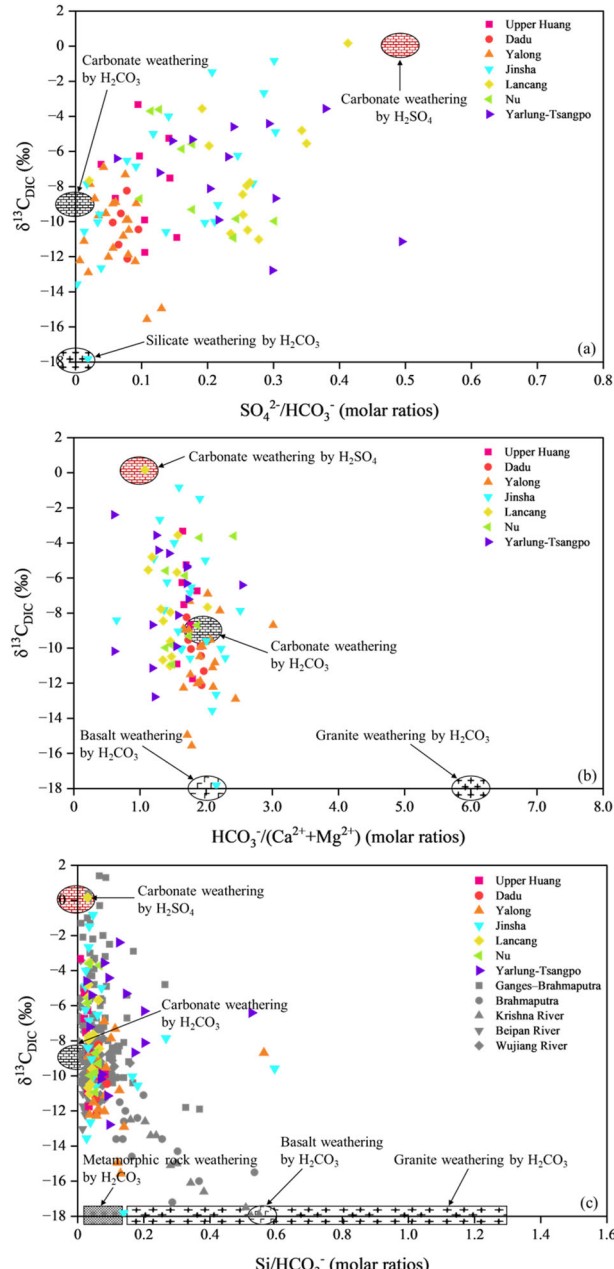

**Fig. 4 | Correlations between δ¹³C_DIC and riverine ion ratios.** The molar ratios of (**a**) SO₄/HCO₃; **b** HCO₃/(Ca+Mg); **c** Si/HCO₃, of the large rivers in the Tibetan Plateau. The end-member of rock weathering is referenced to Refs. 16,51,52. The data for the Ganges-Brahmaputra, Brahmaputra, Krishna, Beipan, and Wujiang Rivers are from refs. 26,52–55.

consistently towards the interior plateau basins, dropping to $11.4 \times 10^3$ mol km$^{-2}$ a$^{-1}$ at Nu River basin (Table 1). The Yarlung Tsangpo river basin with the highest altitude located at the inner area of the plateau serves as a weak CO₂ sink with its average net CO₂ consumption rate as low as $1.5 \times 10^3$ mol km$^{-2}$ a$^{-1}$ (Table 1). The decreasing trend of $\Phi CO_{2net}$ is ascribed to the generally moderative SWR accompanied with accelerated carbonate dissolution by sulfuric acid derived from OWP from the east edge to the interior of the plateau. The pattern for SWR and associated CO₂ consumption is assumed to be mainly controlled by climate (temperature and water accessibility) as silicate weathering is more kinetic-limited within the background high erosion rate, while the CO₂-releasing rate by sulfuric acid reaction with carbonate is primarily dominated by erosion rates. It is worth noting that

the Palong Tsangpo river basin, locating at the Eastern Himalayan syntaxis, one of the fastest exhuming regions on Earth since ~10 Ma[56], serves as a CO₂ source with the highest CWR and an average net CO₂ consumption flux at $-0.07 \times 10^9$ mol a$^{-1}$ (Table 1), suggesting intense CO₂ contracting effects by H₂SO₄-carbonate reaction. Based on our calculation, the whole plateau serves as a long-term CO₂ sink with a consumption flux at $33.2 \times 10^9$ mol a$^{-1}$. In contrast, the value for CO₂ sequestration flux is $79.4 \times 10^9$ mol a$^{-1}$ if the sulfuric acid weathering effect is ignored. Therefore, although sulfide oxidation involved weathering in a single catchment in the Tibetan Plateau have been reported either net CO₂ source or sink[16,49,50], which is all reasonable, depending on their sulfide oxidation rates and the proportion of carbonate and silicate rock exposure, chemical weathering is still an important net CO₂ sink from the comprehensive results of the large river basins in the Tibetan Plateau as a whole. According to our estimation, the sulfuric acid dissolution of carbonate has counteracted about 58% of the CO₂ consumption flux by SW on the plateau. Our study, based on multiple isotope and solute composition observation for major river systems draining in the plateau, for the first time gains the accurate estimation of net CO₂ consumption fluxes of the whole Tibetan Plateau by chemical weathering considering the involvement of sulfuric acid.

The refined CW and SW rates and associated CO₂ budgets calculations in our study show that: (1) the sustained addition of sulfuric acid from OWP could exert a comparable magnitude of CO₂ source flux in the Tibetan Plateau as SW consumption, modified the net CO₂ budgets of plateau weathering. Our work quantitatively shows that the plateau formation of Himalayan orogenesis releases and consumes CO₂ at the same magnitude by weathering and might lead to a weaker disturbance on atmospheric-ocean CO₂ reservoir than previously thought. Therefore, the strong climate-driven feedback needed to offset the orogenic perturbations on the global carbon cycle could be relaxed and should be reconsidered in both modeling and geological record studies. However, attention should be paid to the sampling limitation. First, the ideal procedure is to do the calculation with high-frequency samples through a whole hydrological year, which is challenging due to field logistics and the lack of discharge data in the Tibetan Plateau. The samples in this study are from the high flow season, and average annual discharge data has been used for the flux estimation. Based on the daily dataset obtained in Nujiang[22], it has been estimated that the silicate weathering flux calculated from high flow season sampling is on average 48% lower than the result calculated by the time serials dataset. Therefore, the silicate weathering flux and the associated CO₂ consumption calculation are assumed to be a lower limit. Meanwhile, the sulfuric acid involvement is recognized highest in the high flow season within a hydrology year[22]. Therefore, the flux estimation in this work represents a lower limit of CO₂ sequestration capability for the plateau weathering. Second, it is worthwhile to notice that the weathering signals and CO₂ budgets observed and calculated in our study may represent a non-steady state resulted by increased OWP due to continental glaciations in the Pleistocene and intense glacial activity in the background of modern global change[57], which would highlight the OWP and its effect on chemical weathering[10,16], as thus it is a maximum estimation of H₂SO₄ disturbing in weathering and CO₂ effect in the Tibetan Plateau weathering during its formation. Considering its tectonic and glacial activity variation, the long-term significance of H₂SO₄ could be less, and more work is needed to understand the situation in the deep time; (2) weathering sensitivities of different rock type to landscape variation[15] are quantitatively illustrated in our study from the perspective of large river hydro-geochemistry at continent scale. The orogenesis and plateau formation exert a significantly discriminatory impact on carbonate and silicate weathering, as well as their associated CO₂ effect. CW is significantly enhanced in the plateau, while SW is not anomalous. The formation of the Himalayas Mountain range and the

Tibetan Plateau supplies the largest river runoff volume and sediment fluxes globally[58] on the one hand and serves as a critical factor for the formation of strong South Asian monsoon on the other hand[59–61], both of which will facilitate the subsequent SW in the lower reach for the sediments transported from the plateau by the river networks originated from the plateau. Thus, the $CO_2$ effect of CW and SW should be considered separately for the large river basins spanning orogenic areas and alluvial plains when exploring and modeling the global carbon cycling of the orogenic events in the Cenozoic. The findings of this work propose that different sensitivities of silicate and carbonate weathering mediated by carbonate and non-carbonic acid in different landscapes need to be considered in current or future geologic carbon cycling models as they will bring about significant $CO_2$ sink and source transition of chemical weathering at continental scale, and thus modulate the atmospheric $CO_2$ effects of plateau uplifting event in Cenozoic era.

## Methods

### Sampling and measurements of river water samples

River water samples were collected near the middle of a channel from bridges using high-density polyethylene containers and filtered through a 0.22 μm Millipore membrane filter. One portion of filtrate was acidified to pH <2 with 6 M double sub-boiling distilled $HNO_3$ for cation, trace elements, and Sr isotopic ratio ($^{87}Sr/^{86}Sr$) analysis, and another portion was stored for anion analysis. All containers were previously washed with high-purity $HNO_3$, rinsed with Milli-Q 18.2 MΩ water, and then rinsed with filtrate three times in the field. Samples for dissolved inorganic carbon isotope ($\delta^{13}C$) measurements were preserved with $HgCl_2$ to prevent biological activity in 50 ml polyethylene bottles with air-tight caps and were refrigerated until analysis. For sulfur and oxygen isotopic analysis of $SO_4^{2-}$, the dissolved sulfate in the filtered water samples was precipitated as $BaSO_4$ by adding excess amount of $BaCl_2$ solution after the water was filtered and acidified to a pH value of ~2. This precipitation was filtered, washed, and dried in the laboratory for isotopic analysis.

The temperature (T), pH, and electrical conductivity (EC) were measured with a portable EC/pH meter. The $HCO_3^-$ was determined in the field using the titration method. Anions ($F^-$, $Cl^-$, $SO_4^{2-}$, and $NO_3^-$) were measured by the ion chromatography (Dionex 120) with a precision of ±5%. Cations ($K^+$, $Na^+$, $Ca^{2+}$, and $Mg^{2+}$) were determined by an Atomic Absorption Spectrometry with a precision of ±5%. Dissolved silica ($SiO_2$) concentrations were determined by spectrophotometry using the molybdate blue method, with a precision of ±5%.

The $^{87}Sr/^{86}Sr$ was measured by a Multiple Collector Inductively Coupled Plasma Mass Spectrometry (MC-ICP-MS) after purifying using AG 50W-X8 cation exchange resin in the clean laboratory. The total procedure blank was ~100 pg Sr. The value of the NBS987 standard was 0.710232 ± 0.000020.

For the stable carbon isotope composition of river-dissolved inorganic carbon (DIC) measurements, the filtered samples were injected into glass bottles with phosphoric acid. The $CO_2$ was then extracted and cryogenically purified using a high vacuum line. The $\delta^{13}C$ isotopic ratios were analyzed on Finnigen MAT-252 stable isotope mass spectrometer and the results were expressed relative to VPDB (Vienna Pee Dee Belemnite) standard as $\delta^{13}C_{DIC} = (R_{sample}/R_{standard} - 1) \times 1000$, where $\delta^{13}C_{DIC}$ is the inorganic carbon isotope composition, and $R_{sample}$ and $R_{standard}$ are the $^{13}C/^{12}C$ ratio in water sample and standard, respectively. The analyzing precision for international standard NBS-19 and the samples is better than 0.2‰.

$\delta^{34}S_{SO4}$ and $\delta^{18}O_{SO4}$ for dissolved $SO_4^{2-}$ were determined using the elemental analysis-isotope ratio mass spectrometry (EA-IRMS) and were reported in the δ notation relative to the Vienna Canyon Diablo Troilite reference (V-CDT) and Vienna Standard Mean Ocean Water reference (V-SMOW) in permil, respectively. The analytical precision for $\delta^{34}S_{SO4}$ and $\delta^{18}O_{SO4}$ values of NBS127 and samples was generally better than 0.2‰ and 0.5‰, respectively.

### Calculations for the $CO_2$ consumption rates by silicate and carbonate weathering considering the sulfuric acid involvement

The participation of sulfuric acid involved in carbonate and silicate weathering could be calculated by the simultaneous equations of concentration and isotopic composition of sulfate from different sources (Eqs. 1–3), with the assumption that silicate and carbonate weathering by sulfuric acid are in the same ratio as they are weathered by carbonic acid[44].

$$[SO_4]_{pyrite} = [SO_4]_{SCW} + [SO_4]_{SSW} \tag{1}$$

$$[SO_4]_{SCW} = [SO_4]_{pyrite}/(A+1) \tag{2}$$

$$A = \left(2^*(Ca+Mg)_{sil} + K_{sil} + Na_{sil}\right)/\left(2^*(Ca+Mg)_{carb}\right) \tag{3}$$

$[SO_4]_{SCW}$ and $[SO_4]_{SSW}$ represent the fractions of sulfuric acid consumed by carbonate and silicate weathering, respectively. A is the ratio of carbonic acid consumed by silicate weathering to carbonic acid consumed by carbonate weathering, which could be calculated with the cation partition between different end-members gained in section Supplementary Text Information B. $[SO_4]_{pyrite}$ has been calculated in section Supplementary Text Information B.

The long-term net $CO_2$ consumption budget considering the sulfuric acid involvement in the plateau could be calculated by deducting the $CO_2$ releasing fluxes in carbonate weathering by sulfuric acid from the $CO_2$ consumption fluxes by silicate weathering (Eq. 4) based on the riverine solute sourcing in Supplementary Text Information B and the fractions of sulfuric acid consumed by carbonate and silicate weathering calculated above:

$$\Phi CO_{2net} = \Phi CO_{2sil}^* - \Phi CO_{2SCW}$$
$$= \left((Na+K+2Ca+2Mg)_{sil} - 2[SO_4]_{SSW} - [SO_4]_{SCW}\right) \times discharge/area \tag{4}$$

$\Phi CO_{2sil}^*$ and $\Phi CO_{2SCW}$ indicate yield (mol km$^{-2}$ a$^{-1}$) of atmospheric $CO_2$ consumption/releasing of silicate weathering and carbonate weathering, respectively, with consideration of sulfuric acid involvements. The subscripts of "sil" denote cations originating from silicate weathering.

## Data availability

All data needed to evaluate the conclusion in the paper are present in the paper and the Supplementary Materials. Source data generated in this study have been deposited in the Figshare database (https://doi.org/10.6084/m9.figshare.28092671). Source data are provided with this paper. The digital elevation model (DEM) used in this study is derived from the Shuttle Radar Topography Mission (SRTM) 3 arc-second (90 m) dataset[62], openly available from NASA's Earthdata portal (https://earthdata.nasa.gov/).

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

## Acknowledgements

This work was supported by the National Key Research and Development Program of China (No. 2020YFA0607700), the National Natural Science Foundation of China (Nos. 42422303, 42488201 and 41730857), and the Key Research Program of the Institute of Geology & Geophysics, CAS (No. IGGCAS–202204, 202201). Wenjing Liu acknowledges support from the Youth Innovation Promotion Association CAS (Y2023014).

## Author contributions

W.L., Z.X., and Z.G. designed the project and investigation; W.L., H.S., M.Z., and Y.X. conducted the field work and data analysis and performed the research; W.L. drafted the original manuscript; Z.X., H.S., M.Z., and Z.G. edited and revised the draft. All the coauthors contributed new information and interpretations of the findings during the manuscript writing process.

## Competing interests

The authors declare no competing interests.
