## [Transparent Peer Review file · Nature Communications]

Refined weathering CO₂ budget of the Tibetan Plateau strongly modulated by sulphide oxidation

Corresponding Author: Professor Zhifang Xu

Version 0:

Reviewer comments:

Reviewer #1

(Remarks to the Author)

The manuscript by Liu et al. addresses a critical and long debated problem in continent weathering and carbon cycling. The Tibetan Plateau is the most typical and representative product of Himalaya orogeny in the Cenozoic era. Chemical weathering of the Tibetan Plateau has been regarded leading to dramatic disturbance on atmospheric CO₂ level, global biogeochemistry cycling and climate change on the earth for decades in geoscience community. Very recent studies (e.g., Bufe et al., 2021, NG; Torres et al., 2016, PNAS) found sulfuric acid might played an important role on chemical weathering CO₂ fluxes in high erosion area, such as glacial environment and Taiwan orogeny. However, precise quantitative estimation of H₂SO₄ impact on chemical weathering and related CO₂ budgets constrained by multiple isotope systems is limited, particularly, related estimation and controlling mechanism study is missing for the Tibetan Plateau, which obstructs the understanding on the atmospheric CO₂ effect and related climate impact of plateau formation.

Such work, especially quantitative estimation is very difficult to be realized as it is the top highest and largest plateau with top class large river networks in the world. What's more, strong heterogeneity of geology and climate complicated both the weathering parent rock and weathering fluids here. This manuscript conducted multiple isotopic investigations on river solutes of all the major river systems (including their main channel and main tributaries) draining on the Tibetan Plateau, which gives the opportunity to realize the precise partition of plateau river solute sourcing from different acid-rock reacting systems, and thus the estimation of CO₂ source and sink effect of chemical weathering in these systems in the plateau. It is calculated that nearly half of the CO₂ flux consumed by silicate weathering on the plateau has been counteracted by CO₂ release from carbonate weathering with H₂SO₄ (originated from sulfide oxidation). To my knowledge, this is the first quantitative and systematic calculation on contribution of sulfide oxidation originated sulfuric acid and its disturbance on weathering of the Tibetan Plateau as a whole, thus the net CO₂ budget of chemical weathering of the plateau could be obtained. Another new contribution of this work is the argument that plateau uplifting poses significantly enhanced carbonate weathering but negligible enhancement on silicate weathering, which is critical to solve the negative feedback puzzle of atmospheric CO₂ budget in Cenozoic.

To sum up, I think the topic of the MS is well within the earth and environment scope of NC. Reliable hydro-geochemistry data and isotopic methods are employed for the interpretation. The new findings and arguments have been well presented in the MS. I would recommend the publication of the submitted manuscript on NC after a minor revision.

Other than the general comments above, I also have some specific concerns and suggestions which need to be fully addressed in a moderate revision. Pls find them below:

1. Different kinds of exogenous acids exist in the earth surface. Except for sulfuric acid, another important strong non-carbonic acid agent in a basin is HNO₃. How is the situation of N deposition and its impact on chemical weathering of the plateau? Will the participation of HNO₃ also modulate the basin weathering to a large extent?
2. Line 70-71, discussion need to be strengthened for this argument.
3. Line 77-78, pls specify the reasons of the argument of an increasing portion of OWP originated riverine sulfate.
4. Line 84, what does the pyrite mean? It is necessary to be fully defined when it is mentioned the first time in the MS.
5. Line 111-114, These sentences are kind of confusing to me. To my understanding, the authors mean that the SW flux of the studied river basins is 0.71% of the global SW flux, and their area percentage is 1% of the corresponding global river basin area. While the two value is 10.85% and 6.6% respectively, when the downstream basin area is considered. If it is correct, the sentences need to be rephrased to make it more readable. Furthermore, does that mean downstream of the large rivers originated from the Tibetan Plateau have higher chemical weathering fluxes of silicate weathering? Is it the lithology or

the climate dominating this phenomena?

6. It was mentioned that the area-averaged weathering rate of the Tibetan Plateau is calculated at $41.7 \text{ t km}^{-2} \text{ a}^{-1}$ in Line 107, doubled of the global value. However, the authors argued later in the Line 116 that the SW rate in the Tibetan Plateau is not enhanced as previous thought. Does that implicate a much stronger accelerating of CW on the plateau? As I point out in the above comment, SW will be largely accelerated downstream when the rivers flowing into the plain, what's the situation of carbonate weathering?

7. Line 129-133, why silicate weathering in Palong Tsangpo river basin has not been largely enhanced by high erosion rate there as carbonate weathering.

8. Line 149-151, it is argued that the major river basins take up to 1% of the basin area of the global value, but contributed 4.17% of the OWP fluxes. I think it would be necessary to explicate the reasons and explore its implications.

9. I suggest that some contents in the Supporting Text Information need to be removed to the main text to make the MS more easily follow for readers. e.g., I think it would be better to put the content of "Quantification of Riverine Sulfate Source Contribution" and "Calculations for the CO₂ Consumption Rates by Silicate and Carbonate Weathering Considering the Sulfuric Acid Involvement" to the method section.

10. Similarly, some figure in the SI should be removed to the main body of the MS, e.g. Fig. S3 is actually very important as it is the basic of the chemical weathering rates calculation and the quantitative estimation of H₂SO₄ involvement in chemical weathering.

11. Line 180-182, pls rephrase this sentence to make it more readable.

12. I would recommend a language polishing of the MS by an English native speaker.

Reviewer #2

(Remarks to the Author)

Please see the attachment for my comments.

Reviewer #3

(Remarks to the Author)

In this manuscript, Liu and colleagues present a comprehensive dataset on stream water chemistry in the Tibetan Plateau, with a focus on sulfate flux. Using several mixing models, referred to as forward models, the authors quantify the flux of silicate and carbonate weathering driven by CO₂ and pyrite oxidation. The main conclusions are:

1. The area-averaged silicate weathering flux is slightly lower than the global mean, while the area-averaged carbonate weathering flux is twice the global mean.

2. The acidity generated through pyrite oxidation contributes to approximately 1/5 of the carbonate weathering flux, potentially reducing the CO₂ consumption flux through silicate weathering by about 60%.

There is no doubt that the authors have collected a valuable dataset, which could greatly benefit the geological community once published. However, I have some concerns about the manuscript:

Sampling Conditions: Based on the map in Fig. 1, the sampling locations cover the Tibetan Plateau well, which is commendable. However, it is unclear when and under which hydrological conditions (e.g., discharge) the samples were collected. Without this information, it is difficult to determine how the fluxes were calculated. Different discharge-concentration relationships could lead to significantly different flux values, making the flux values representing the whole basin questionable.

Mechanistic Understanding: The manuscript lacks sufficient interpretation of the observed patterns. For instance, to what extent can the variation in weathering fluxes among basins be explained by differences in erosion/uplift rates, climate, or lithology? More detailed discussion on these aspects is needed to enhance the mechanistic understanding.

Mixing Model Constraints: The mixing models (forward models) are not well constrained. Their validity largely depends on the accuracy of the endmembers. The uncertainties of the endmembers (e.g., Ca/Na, Mg/Na ratios, sulfur isotopic compositions) should be accounted for to estimate the uncertainties of the results (fraction of endmembers).

Version 1:

Reviewer comments:

Reviewer #1

(Remarks to the Author)

The authors replied to the comments accurately. I think it can be accepted.

Reviewer #2

(Remarks to the Author)

After reading the revision, I have no further comments and suggest to accept the manuscript.

Reviewer #3

(Remarks to the Author)

I believe the authors have not fully addressed the issue of solute flux estimation. It is difficult to accept that the concentrations from samples collected at a specific time of the year can represent the average values for the entire year. While I understand that it may be challenging to obtain multiple samples in Tibet, the manuscript should mention this sampling limitation and discuss how it may impact the flux calculation.

Point-by-point response to the reviewers' comments

Reviewer #1 (Remarks to the Author):

The manuscript by Liu et al. addresses a critical and long debated problem in continent weathering and carbon cycling. The Tibetan Plateau is the most typical and representative product of Himalaya orogeny in the Cenozoic era. Chemical weathering of the Tibetan Plateau has been regarded leading to dramatic disturbance on atmospheric CO₂ level, global biogeochemistry cycling and climate change on the earth for decades in geoscience community. Very recent studies (e.g. Bufe et al., 2021, NG; Torres et al., 2016, PNAS) found sulfuric acid might played an important role on chemical weathering CO₂ fluxes in high erosion area, such as glacial environment and Taiwan orogeny. However, precise quantitative estimation of H₂SO₄ impact on chemical weathering and related CO₂ budgets constrained by multiple isotope systems is limited, particularly, related estimation and controlling mechanism study is missing for the Tibetan Plateau, which obstructs the understanding on the atmospheric CO₂ effect and related climate impact of plateau formation.

Such work, especially quantitative estimation is very difficult to be realized as it is the top highest and largest plateau with top class large river networks in the world. What's more, strong heterogeneity of geology and climate complicated both the weathering parent rock and weathering fluids here. This manuscript conducted multiple isotopic investigations on river solutes of all the major river systems (including their main channel and main tributaries) draining on the Tibetan Plateau, which gives the opportunity to realize the precise partition of plateau river solute sourcing from different acid-rock reacting systems, and thus the estimation of CO₂ source and sink effect of chemical weathering in these systems in the plateau. It is calculated that nearly half of the CO₂ flux consumed by silicate weathering on the plateau has been counteracted by CO₂ release from carbonate weathering with H₂SO₄ (originated from sulfide oxidation). To my knowledge, this is the first quantitative and systematic calculation on contribution of sulfide oxidation originated sulfuric acid and its disturbance on weathering of the Tibetan Plateau as a whole, thus the net CO₂ budget of chemical weathering of the plateau could be obtained. Another new contribution of this work is the argument that plateau uplifting poses significantly enhanced carbonate weathering but negligible enhancement on silicate weathering, which is critical to solve the negative feedback puzzle of atmospheric CO₂ budget in Cenozoic.

To sum up, I think the topic of the MS is well within the earth and environment scope of NC. Reliable hydro-geochemistry data and isotopic methods are employed for the interpretation. The new findings and arguments have been well presented in the MS. I would recommend the publication of the submitted manuscript on NC after a minor revision.

Other than the general comments above, I also have some specific concerns and suggestions which need to be fully addressed in a moderate revision. Pls find

them below:

1. Different kinds of exogenous acids exist in the earth surface. Except for sulfuric acid, another important strong non-carbonic acid agent in a basin is HNO₃. How is the situation of N deposition and its impact on chemical weathering of the plateau? Will the participation of HNO₃ also modulate the basin weathering to a large extent?

Response: Yes, we agree that N deposition might be an important source of exogenous acids in a river basin, especially for those with intense atmospheric N deposition. However, its input in the Tibetan Plateau should be minor. The plateau is one of the most remote and pristine area in the world. Previous work has been conducted for the Yarlung Tsangpo River Basin on the plateau to evaluate the N deposition situation in the basin (Liu et al., 2022, JH). It was calculated that the atmospheric N deposition accounted for only ~5% of the riverine NO₃⁻ in the Yarlung Tsangpo River. Besides, the plateau rivers have much lower NO₃⁻ concentration range than global major rivers (**Table S3**. A table compiling the nitrate concentration and isotope characteristics of the Yarlung Tsangpo River and global rivers, in Liu et al., 2022, WR). So, the atmospheric N deposition impact on chemical weathering was assumed negligible in the plateau.

Liu, W., et al. (2022). "Driving forces of nitrogen cycling and the climate feedback loops in the Yarlung Tsangpo River Basin, the highest-altitude large river basin in the world." Journal of Hydrology 610: 127974.

Liu, W., et al. (2022). "Time-series monitoring of river hydrochemistry and multiple isotope signals in the Yarlung Tsangpo River reveals a hydrological domination of fluvial nitrate fluxes in the Tibetan Plateau." Water Research 225: 119098.

2. Line 70–71, discussion need to be strengthened for this argument.

Response: Thanks for your suggestion. The plots of ⁸⁷Sr/⁸⁶Sr and cation ratio index on Fig S3 for endmember identification show that most samples distribute more closer to carbonate dissolution endmembers, especially for the mainstream samples. Here in the very beginning of the first section of “results and discussions”, the hydro-geochemistry composition and the Sr isotope data are employed to qualitatively get a general picture of weathering solute origin. More quantitative estimations on solute origine are given in the followed-up sections. Interpretations and details about hydro-chemistry and Sr isotope implications on this argument have been added and the logic has been streamlined in the revision in Line 73~79.

3. Line 77–78, pls specify the reasons of the argument of an increasing portion of OWP originated riverine sulfate.

Response: Rainwater, gypsum dissolution and oxidation of pyrite are the main contributor of riverine sulfate. The former two have positive $\delta^{34}\text{S}_{\text{SO}_4}$ values normally around 5‰ and higher than 10‰ as reported in previous studies. While the oxidation of pyrite normally has negative $\delta^{34}\text{S}_{\text{SO}_4}$ values. Detailed endmember confirmation process could be found in the “Quantification of Riverine Sulfate Source Contribution” section in the supplementary information section. So, the decreasing trend of $\delta^{34}\text{S}_{\text{SO}_4}$

down the mainstream of the studied rivers (Table S1) implicates an increasing input from the endmember with lower $\delta^{34}\text{S}_{\text{SO}_4}$ value than the river water, suggesting an increasing contribution from OWP with negative $\delta^{34}\text{S}_{\text{SO}_4}$. Line 85~87 in the revision.

4. Line 84, what does the f_{pyrite} mean? It is necessary to be fully defined when it is mentioned the first time in the MS.

Response: Thanks for the reminding. f is the proportion of the sulfate derived from rainwater, gypsum dissolution and oxidation of pyrite. So, f_{pyrite} noted the fraction of riverine SO_4^{2-} originated from oxidation weathering of pyrite. It has been clarified in Line 111 in the revised manuscript.

5. Line 111~114, These sentences are kind of confusing to me. To my understanding, the authors mean that the SW flux of the studied river basins is 0.71% of the global SW flux, and their area percentage is 1% of the corresponding global river basin area. While the two value is 10.85% and 6.6% respectively, when the downstream basin area is considered. If it is correct, the sentences need to be rephrased to make it more readable. Furthermore, does that mean downstream of the large rivers originated from the Tibetan Plateau have higher chemical weathering fluxes of silicate weathering? Is it the lithology or the climate dominating this phenomena?

Response: Sorry for the confusion. Yes, this is what we mean in the manuscript. The statements have been rephrased to make it more readable and the comparison has been specified in the revised manuscript in Line 123~126. We attribute the enhancement of silicate weathering to the longer residence time and monsoonal climate in the flood plain downstream of the large rivers originated from the plateau. Pls find the details in Line 129~137 in the revised MS.

6. It was mentioned that the area-averaged weathering rate of the Tibetan Plateau is calculated at $41.7 \text{ t km}^{-2} \text{ a}^{-1}$ in Line 107, doubled of the global value. However, the authors argued later in the Line 116 that the SW rate in the Tibetan Plateau is not enhanced as previous thought. Does that implicate a much stronger accelerating of CW on the plateau? As I point out in the above comment, SW will be largely accelerated downstream when the rivers flowing into the plain, what's the situation of carbonate weathering?

Response: Thanks for the insightful comments. Yes, the calculated total weathering rate of TP is twice of the global value, but SWR in the Tibetan Plateau is not enhanced as previous thought. It is generally the same level of the global value despite the high erosion rate there, while carbonate weathering has been accelerated compared with the global mean, although the plateau has much lower mean annual temperature in the plateau. The high erosion rate is assumed as the first order of controlling factors accelerating the carbonate weathering there considering its fast-reacting rate with weathering fluids. But the promoting of silicate weathering reactions is relatively limited for its much slower weathering rates (normally 10^3 orders lower than carbonate). Carbonate weathering is also enhanced in the flood plain area, but the main controlling

factor is the high temperature and water accessibility there due to the monsoonal climate.

7. Line 129–133, why silicate weathering in Palong Tsangpo river basin has not been largely enhanced by high erosion rate there as carbonate weathering.

Response: Palong Tsangpo river basin is located at the Eastern Himalayan syntaxis, a hotspot for high erosion rate on the Earth surface. It is a typical alpine valley area with abundant rainfall and widespread marine glaciers. Palong Tsangpo river basin is characterized by strong tectonic and glacier activity, intense precipitation erosion but low temperature (MAT at 8.6 °C, Zeng et al., 2019; Zhang et al., 2023). The above special combination of erosion and climate background promotes the obvious high carbonate weathering rates, but the enhancement on silicate weathering rates are relatively limited for short water residence time and low temperature there. In another word, silicate weathering is intensely kinetic-limited there.

Zeng, X. et al. Characteristics and Geneses of low frequency debris flow along ParlungZangbo River Zone-take ChaoBulongba gully as an example. Sci. Tech. Eng. 19, 103-107 (2019).

Zhang, G. The controlling effect of structure and climate on the distribution of debris flow in Palong Zangbu Basin 1-69 (Xizang Univ., 2023).

8. Line 149–151, it is argued that the major river basins take up to 1% of the basin area of the global value, but contributed 4.17% of the OWP fluxes. I think it would be necessary to explicate the reasons and explore its implications.

Response: Thanks for your suggestion. Physical erosion rate is the first-order controlling factor for oxidation weathering of pyrite. As a region with intense tectonic activity and the widest distribution of low-latitude glaciers, the Tibetan Plateau serves as one of the hotspots for pyrite exposure, which lead to the disproportionately high OWP originated sulfate flux. Related explanation was added in the revised manuscript in Line 170~172.

9. I suggest that some contents in the Supporting Text Information need to be removed to the main text to make the MS more easily follow for readers. e.g., I think it would be better to put the content of “Quantification of Riverine Sulfate Source Contribution” and “Calculations for the CO₂ Consumption Rates by Silicate and Carbonate Weathering Considering the Sulfuric Acid Involvement” to the method section.

Response: Thanks for your suggestion. Yes, we agree to remove “Calculations for the CO₂ Consumption Rates by Silicate and Carbonate Weathering Considering the Sulfuric Acid Involvement” to the method section in the revision in Line 270~293 from the SI. As for the “Quantification of Riverine Sulfate Source Contribution” section, it contains more discussion on the endmember investigation and confirmation, which is the premise of contribution calculation, the results of riverine SO₄²⁻ contribution and their implications, instead of only the procedure of riverine sulfate source calculation. So, we would like to leave this section in the Supporting Text Information.

10. Similarly, some figure in the SI should be removed to the main body of the MS, e.g. Fig. S3 is actually very important as it is the basic of the chemical weathering rates calculation and the quantitative estimation of H₂SO₄ involvement in chemical weathering.

Response: Thanks for your suggestion. The original Fig. S3 has been removed to the main body of the revised MS to illustrate the solute source contributions from different endmembers, which is the new Fig 2 in the revision.

11. Line 180–182, pls rephrase this sentence to make it more readable.

Response: Sorry for the confusion. This sentence has been revised in the revision in Line 216~218.

12. I would recommend a language polishing of the MS by an English native speaker.

Response: Thanks for your suggestion. The language of the revision has been polished.

Reviewer #2 (Remarks to the Author):

Please see the attachment for my comments.

Dear Editor,

I have read through the manuscript entitled “Refined weathering CO₂ budget of the Tibetan Plateau strongly modulated by sulphide oxidation” by Liu et al. This paper reported refined sulfide oxidation weathering flux with carbonate and silicate rocks. The biggest challenge for this theme in my opinion is that previous studies used model calculation by setting the majority of SO₄ sourced from sulfides, and therefore unable to accurately isolate the pyrite weathering flux, due to unavailable S isotopic data for many river basins (e.g., Torres et al., 2017, PNAS). Liu et al. used C-S isotopes for the first time accurately constrained the sulfide oxidation flux and CO₂ consumption in the entire Tibetan Plateau (TP), and finally found that the TP is an important carbon sink. This is a valuable contribution for understanding the carbon source and sink in the entire TP, as individual river basin in this region showed either a net source or sink. Their findings are particularly useful to assess the carbon cycle processes and fluxes in the tectonically active TP, a hot point region that has long been linked to tectonic uplift and the Cenozoic cooling.

The manuscript is generally well-written. The data support the conclusions. I have made some comments for the data interpretation that I think the authors can be solved at current version. Overall, I recommend the publication of this manuscript after minor modifications.

Comments:

Line 36: Change the “glacial” to “glaciers”

Response: Revised in line 36.

Line 70–71: “Both the river hydro-chemistry and Sr geochemistry suggest a highlighted carbonate dominated weathering regime on the plateau.” Why the Sr geochemistry suggest a carbonate dominated weathering? Are there metamorphic

carbonate? The average $^{87}\text{Sr}/^{88}\text{Sr}$ for the TP is 0.71245, which is much higher than the limestone 0.709 or global evaporite 0.710–0.711. I agree with the hydro-chemistry interpretation that showing Ca–Mg– HCO_3^- dominated ionic compositions, but I can not follow the $^{87}\text{Sr}/^{88}\text{Sr}$ statement. I suggest the authors to clarify it here for Sr isotopic description.

Response: Thanks for your suggestion. Metamorphic carbonate originated high radiogenic Sr isotope has been well documented in the studies in southern slope of the plateau (Fig S2), but has not been founded on the plateau river basins. Huh argued that the analyses of data for the Himalayas and the Tibetan Plateau suggest that the high radiogenic Sr feature is localized to the Himalayan front draining the High Himalaya Crystalline (HHC) and Lesser Himalaya (LH) and has not been observed in other peripheral regions of the Tibetan Plateau or in the two syntaxes (Huh, 2010), although the $^{87}\text{Sr}/^{86}\text{Sr}$ ratios are highly variable for tributaries in individual river basin. To avoid the heterogeneity and gain a general understanding at the whole plateau scale, we look at the main channel data, and the average $^{87}\text{Sr}/^{86}\text{Sr}$ for mainstream of large rivers in the TP is 0.7111, which is slightly higher than the value range of carbonate and evaporite endmember, but significantly lower than well documented normal silicate endmember value range (0.72~0.73, Oliver et.al., 2003; Wang et.al., 2007; Chetelat et.al., 2008). In addition, the plots of $^{87}\text{Sr}/^{86}\text{Sr}$ and cation ratio index on Fig S4 showed that most samples are distributed more closer to carbonate dissolution endmembers. In the very beginning of the first section of “results and discussions”, the hydro-geochemistry composition and the Sr isotope data here are employed to qualitatively get a general picture of weathering solute origin. More interpretations and details about hydro-chemistry and Sr isotope implications on carbonate weathering dominated weathering regime have been added in this section and the logic has been streamlined in the revision in Line 73~79.

Wang Z.-L., Zhang J. and Liu C.-Q. (2007) Strontium isotopic compositions of dissolved and suspended loads from the main channel of the Yangtze River. Chemosphere 69(7), 1081–1088.

Chetelat, B., et al. (2008). "Geochemistry of the dissolved load of the Changjiang Basin rivers: Anthropogenic impacts and chemical weathering." Geochimica et Cosmochimica Acta 72(17): 4254-4277.

Huh, Y. (2010). "Estimation of atmospheric CO_2 uptake by silicate weathering in the Himalayas and the Tibetan Plateau: a review of existing fluvial geochemical data." Geological Society, London, Special Publications 342(1): 129-151.

Oliver L., Harris N., Bickle M., Chapman H., Dise N. and Horstwood M. (2003) Silicate weathering rates decoupled from the $^{87}\text{Sr}/^{86}\text{Sr}$ ratio of the dissolved load during Himalayan erosion. Chem. Geol. 201(1–2), 119–139.

Line 81–85: I suggest to move these two sentences “Riverine sulfate origins are calculated with SO_4^{2-} concentration and $\delta^{34}\text{S}_{\text{SO}_4}$ value...higher f_{pyrite} for the studied river system indicates significant sulfuric acid involvements in the

plateau weathering.” to after the Line 103 or other sections with sulfide weathering discussions. The authors started the quantified calculation regarding carbonate and silicate weathering contribution from Line 86–103, which provided basic information for endmember input. At this basis, then continue to calculate the sulfide contribution, and discuss the importance of sulfuric acid involvement weathering by sulfide oxidation.

Response: Thanks for your comment. Yes, we agree that the suggestions will help to streamline the logic. The content has been reorganized in Line 105 to Line 113 in the revision.

Line 91–92: This is a little bit unclear for general readers. You may clarify this as “...are more complicated than rivers in other global geomorphology units, because the TP are characterized by almost highest uplift rates, widespread mid-latitude glaciers, contrasting climate (across from >2000 mm/yr precipitation in southern Himalayan to extreme drought in plateau inland regions) and vegetations, and complex lithologies, particularly the meta-carbonate weathering.

Response: Thanks for your suggestion. Considering the followed-up comment from the review 2# (Line 146–148: This sentence “Partitioning the riverine sulfate sources... as a whole” is similar from in the line 90–92 “However, the river ...as revealed above”. I suggest to merge these two places in somewhere.), we have reorganized these contents in Line 93~103 in the revised manuscript, and the knowledge gap on solute identification for the TP rivers are clarified in the introduction section (Line 45~49).

Line 94–96: Yes, this is novel to assess the sulfide weathering by systematically adding new sulfide endmember, so that can better constrain their contributions in the whole Tibetan Plateau.

Response: Thanks for your recognition. One of our motivations of this work to quantifying the sulfuric acid disturbance in the plateau weathering process. It is critical to constrain the OWP endmember and contributions to the riverine sulfate there.

Line 107: Change to “the area-averaged total chemical weathering rate...”, so that here is clear not regarding physical weathering rate.

Response: Thanks for your suggestion. Yes, we mean “total chemical weathering rate” here. Revised in the revision in Line 118.

Line 113: Change “at” to “extending to 6.6%”.

Response: Thanks for your suggestion. “extending” here is indeed a better expression. We have revised the content of this sentence in Line 123~125 in the revision to make it more readable.

Line 114: Adding “A large spatial comparison indicates that the plateau river basins...”

Response: Thanks for your suggestion. We do need a statement here to make it more easy following for readers to start the continental scale comparison. We have revised

the sentence in Line 125~129 in the revision.

Line 114-125: I think this comparison is very interesting, with which demonstrating that although the highly-eroded upstream plateau produced large amounts of debris with fresh mineral surface, the silicate weathering rates were still much lower than downstream area, suggesting a central role of climate (precipitation, runoff, temperature) together to enhance the silicate weathering rate. This is an additional supplementary to explain the uplift hypothesis.

Response: Thanks for your recognition. This is indeed one of our arguments of this work that the silicate weathering in the plateau is limited and incipient in the plateau region (or the orogeny area) although tectonic and glacier activity promote the debris generating and thus the fresh reactive surface for silicate minerals. The silicate weathering reactions is not enhanced as previous thought as the weathering regime for silicate is kinetic limited for the short water-mineral resident time and low temperature on the plateau. In another aspect, the function of climate controlling on silicate weathering could be seen in the downstream of the large rivers originating from the plateau.

Line 133-136: I agree with the impact from physical erosion, but here I think it maybe just the sulfide oxidation driven by rapid erosion accelerates the carbonate weathering, and finally results in higher CWR. This also agree with your sulfide weathering observations, and then you continue to introduce the CO₂ consumption from Line 137.

Response: Thanks for your suggestion. The statement has been revised to explore the mechanism of accelerated OWP and its functions on carbonate weathering to in Line 149~154 in the revision.

Line 139: It' s better to add references for the glacier activity and erosion and OWP, as this is the first time to mention the glacier effects on OWP in the main text. You may cite Cao et al., 2023, 901, 165842STOTEN. Their paper reported extensive sulfide oxidation in a typical glacial catchment in the NE Tibetan Plateau.

Response: Thanks for your suggestion. Yes, more references do help to illustrate the relationship between glacier activity, erosion and OWP. We did more investigation and have added some representative work here. Pls find them in Line 157 in the revision.

Line 140-143: Yes, this is one of the key evidence that showing most areas are affected by sulfide weathering in the whole Tibetan Plateau region when looking at the $\delta^{13}\text{CDIC}$ values.

Response: Thanks for the comment. The Tibetan plateau is indeed with complicated chemical weathering system considering its bedrock and weathering acid agents. Combining use of multiple isotopic systems would help to gain independent evidence lines to test the hypothesis and make quantitative estimation.

Line 146-148: This sentence “Partitioning the riverine sulfate sources... as a

whole” is similar from in the line 90-92 “However, the river ...as revealed above”. I suggest to merge these two places in somewhere.

Response: Thanks for your suggestion. The statements in Line 90-92 in the previous submission was intended to present the difference of solute identification between the plateau river and rivers in other geomorphology units. After serious considering on the comment and the conciseness of the content, we decided to delete this sentence, and added the statements “The river solute source identification and quantification for large rivers in the Tibetan Plateau are not well constrained as significant variations exist on weathering bedrock and weathering acid agents, and systematic dataset to calculate the riverine ion contribution from different weathering reactions are in lacking. Investigation of the fluvial geochemistry is needed to gain quantitative estimation of the chemical weathering and associated CO₂ fluxes for the plateau as a whole.” in the introduction section in Line 45~49 in the revision.

Line 164-166: It’s better to add some discussion after the calculations. The result is very interesting in my opinion. You may say after “...sulfuric acid weathering effect is ignored.” Like this: “Therefore, although sulfide oxidation involved weathering in a single basin in the Tibetan Plateau reported either net CO₂ source or sink (references), which is all reasonable, depending on their sulfide oxidation rates and might also be the propositions of carbonate and silicate rocks, it is still an important net CO₂ sink from the comprehensive results of the entire Tibetan Plateau. We further find that the sulfuric acid dissolution ...”

Response: Thanks for your insightful and construction comment! Yes, we totally agree that more discussion here will help to highlight the findings. We have added more arguments here based on the suggestions. Pls find them in Line 195~199 in the revision.

Line 167-168: Change “Our study...quantifies...” to “Our study, based on measured and compiled $\delta^{34}\text{S}_{\text{SO}_4}$ and solute compositions, for the first time accurately quantifies...”

Response: Thanks for your suggestion. We have revised the statements and pointed the tools of the study in Line 201-204.

Line 172-176: I agree with this because glacial catchment generally has higher erosion rates and thus pyrite exposure and oxidation. It’s better to add a reference saying that glacial basins reported higher erosion and sulfide oxidation, so that can well support your statement here.

Response: Thanks for your suggestion. Several recent studies have been cited and given credits in the revision in Line 210 to support the arguments here.

Line 190-196: I think the biggest contribution of this paper is that the authors are able to quantify the sulfide weathering flux by using C and S isotopes in the tectonically active Tibetan Plateau, which is a challenge for previous studies by using model calculation which can not distinguish sulfide oxidation

and gypsum dissolution due to no available S isotopes for many river basins, and therefore existing large uncertainties (see discussion by Torres et al., 2017, 114, 8716–8721, PNAS, <https://doi.org/10.1073/pnas.1702953114>). This can be add somewhere to highlight your contributions.

Response: Thanks for your recognition. We seriously considered the related comments and suggestions on highlighting the findings of the manuscript. The significance, research motivation and findings of this study have been highlighted several places in the revised manuscript in Line 44~48, Line 103~112, Line 195~204, and also the new version of last paragraph of the main text.

Reviewer #3 (Remarks to the Author):

In this manuscript, Liu and colleagues present a comprehensive dataset on stream water chemistry in the Tibetan Plateau, with a focus on sulfate flux. Using several mixing models, referred to as forward models, the authors quantify the flux of silicate and carbonate weathering driven by CO₂ and pyrite oxidation. The main conclusions are:

1. The area-averaged silicate weathering flux is slightly lower than the global mean, while the area-averaged carbonate weathering flux is twice the global mean.
2. The acidity generated through pyrite oxidation contributes to approximately 1/5 of the carbonate weathering flux, potentially reducing the CO₂ consumption flux through silicate weathering by about 60%.

There is no doubt that the authors have collected a valuable dataset, which could greatly benefit the geological community once published. However, I have some concerns about the manuscript:

Sampling Conditions: Based on the map in Fig. 1, the sampling locations cover the Tibetan Plateau well, which is commendable. However, it is unclear when and under which hydrological conditions (e.g., discharge) the samples were collected. Without this information, it is difficult to determine how the fluxes were calculated. Different discharge-concentration relationships could lead to significantly different flux values, making the flux values representing the whole basin questionable.

Response: Thanks for your reminding. Yes, we agree that both the sampling location and sampling period matters a lot for the flux calculations. The sampling and parameter details for flux calculation have been added in Line 55~59, Line 115~117 in the revision. About the representativeness of flux values, one of the advantages of large river water study is that rivers are natural integrators of different weathering processes and fluxes over their vast drainage areas, which could provide a nature tool averaging the heterogeneity of chemical weathering in a basin. The fluxes and area-averaged yield of chemical weathering and uptake of atmospheric CO₂ of the studied river basin can be calculated for the river basins with the results from the solute source quantification section, the basin area and the average annual discharge data of different hydrology stations on the main channel (hydrology station location and information could be found in Fig. 1 and Table 1 in the revised manuscript.). The hydrology stations used for the final weathering flux calculation for each large river basin are the furthest ones

down the mainstream at the edge of the plateau, which is at the exit of the plateau, representing the weathering fluxes of the river basins on the plateau.

Mechanistic Understanding: The manuscript lacks sufficient interpretation of the observed patterns. For instance, to what extent can the variation in weathering fluxes among basins be explained by differences in erosion/uplift rates, climate, or lithology? More detailed discussion on these aspects is needed to enhance the mechanistic understanding.

Response: Thanks for the insightful comments. Yes, erosion/uplift rates, climate, and lithology play together to control the weathering rate and fluxes, as well as the associated CO₂ fluxes of chemical weathering. The Tibetan Plateau is indeed one of the geomorphology units with most highly varied geology and climate background, which complicates the weathering processes there. This is also the reason for different or even controversial results and findings have been reported previously from different catchments at smaller scale. So, this work was mainly designed from the perspective of all the large river systems draining on the plateau, which provide us an opportunity to gain the chemical weathering fluxes and associated CO₂ budgets for the whole plateau scale. Thus, we can avoid the heterogeneity of weathering conditions to gain accurate estimations of weathering material fluxes of the plateau, as well as an overall evaluation on the role sulfuric acid playing and understanding on the weathering regime of the whole plateau. With these efforts, we can also make comparison between the river drainages on the plateau and downstream in the flood plain. A climate domination could be recognized for silicate weathering in this comparison between the upper reaches and down reaches (Line 129~137). The erosion controlling mechanisms on weathering fluxes are explored in Line 149~154 in the revision. Thanks for this constructive suggestion, the controlling factors on spatial variation of weathering between river basins on the plateau has also been strengthened in the revised manuscript in Line 179~189 to help the mechanistic understanding.

Mixing Model Constraints: The mixing models (forward models) are not well constrained. Their validity largely depends on the accuracy of the endmembers. The uncertainties of the endmembers (e.g., Ca/Na, Mg/Na ratios, sulfur isotopic compositions) should be accounted for to estimate the uncertainties of the results (fraction of endmembers).

Response: Thanks for the constructive comments. The forward model sequentially subtracts components with pre-assigned compositions from the total riverine loads for each element. Yes, the accuracy of the endmembers is critical for the mixing models. The types of endmembers for the river solutes were carefully considered based on the previous studies on basin weathering as well as the specific background of the Tibetan Plateau. The adopted element ratios and sulfur isotopic composition for the endmembers used in the calculation of cationic load contribution and riverine sulfate from different reservoirs are carefully investigated and distinguished by intensely compiling of all the available data published. For the Ca²⁺/Na⁺ and Mg²⁺/Na⁺ ratios which were used to calculate the Ca and Mg contribution from silicate weathering to

the river water, considering the large scale of the studied river basins, it is difficult to select appropriated values from bedrock data due to the complicated lithology exposure in such large scale river basins in this study. So, we tend to use the values of global major river basins reported by Gaillardet et al. (1999), with The $\text{Ca}^{2+}/\text{Na}^{+}$ and $\text{Mg}^{2+}/\text{Na}^{+}$ ratio at 0.35 ± 0.15 of 0.24 ± 0.12 , respectively, which were widely referenced in large river basin scale study in different basin weathering studies at continent scale, and also in the references with which we were making the comparisons. Based on this insightful comment, we have done the uncertainty propagation for the endmember values. The error of the silicate weathering contribution percentage to the cationic loads was calculated considering the $\text{Ca}^{2+}/\text{Na}^{+}$ and $\text{Mg}^{2+}/\text{Na}^{+}$ ratio uncertainty, and the result is between 0.5 to 7.8% (averaging at 1.8%) for the mainstream samples, and between 0.1 to 8.14% (averaging at 1.83%) when including the tributary samples. For the proportion of the sulfate derived from rainwater, gypsum dissolution and oxidation of pyrite, the $\delta^{34}\text{S}_{\text{SO}_4}$ for rainwater, gypsum dissolution and pyrite oxidation are assigned to be $4\pm 2\text{‰}$, $17.5\pm 1.5\text{‰}$ and $-8\pm 3\text{‰}$ (detailed in the supplementary information BIII) and the propagated uncertainties for f_{pyrite} (the percentage of riverine sulfate originated from oxidation weathering of pyrite) are between 4.06 to 10.77%. Considering the calculated contribution ranges, the errors brought about by the endmember value are relatively low, which will not modify our main findings and fluxes estimation.

Gaillardet, J., et al. (1999). "Global silicate weathering and CO₂ consumption rates deduced from the chemistry of large rivers." Chemical Geology 159(1-4): 3-30.

Point-by-point response to the reviewers' comments

REVIEWERS' COMMENTS

Reviewer #1 (Remarks to the Author):

The authors replied to the comments accurately. I think it can be accepted.

Reviewer #2 (Remarks to the Author):

After reading the revision, I have no further comments and suggest to accept the manuscript.

Reviewer #3 (Remarks to the Author):

I believe the authors have not fully addressed the issue of solute flux estimation. It is difficult to accept that the concentrations from samples collected at a specific time of the year can represent the average values for the entire year. While I understand that it may be challenging to obtain multiple samples in Tibet, the manuscript should mention this sampling limitation and discuss how it may impact the flux calculation.

Response: Yes, solute flux estimation is largely dependent on the sampling time and discharge variation. The ideal situation is to do the calculation with high frequency samples through a whole hydrological year (preferably, 40 temporal chemical data points with synchronous discharge from each river are necessary suggested by Moon et al., 2014). However, it is challenging to do such observations for all the large river systems in this work at the same time due to field logistics in the plateau and lacking of discharge data. We use single time sampling strategy and average discharge data for the flux estimation here. The uncertainty could be discussed based on a high-frequency study for a hydrology station at the mainstream of Nujiang we conducted before (Liu et al., 2023). It was estimated that the silicate weathering flux was 7.4×10^9 mol/yr and 15.9×10^9 mol/yr for the high flow (calculated with the daily concentration data from July, August and September) and low flow season (calculated with the daily concentration data from April to June, and from October to November), respectively, averaging at 12.5×10^9 mol/yr for the whole year estimation, when using the average discharge data of the year. While the value is 17.7×10^9 mol/yr and 12×10^9 mol/yr, averaging at 14.3×10^9 mol/yr, when using the daily discharge data. The perfect situation is the latter one, to use daily concentration and discharge data. Based on the Nujiang dataset and recognition, the silicate weathering flux calculated from sampling in the high flow season adopted in this study is estimated to be averagely 48% lower than the value calculated by daily dataset. In all, the silicate weathering flux and the associated

CO₂ consumption calculation is assumed to be a lower limit while the sulfuric acid involvement is an upper limit in this work, leading to a lowest estimation of CO₂ sequestration capability for the plateau weathering. We have clarified the sampling limitation and its impact on the flux calculation in Line 208~ 218.

*Moon, S., et al. (2014). "New estimates of silicate weathering rates and their uncertainties in global rivers." *Geochimica et Cosmochimica Acta* 134(0): 257-274.*

*Liu, W., et al. (2023). "Hydrochemistry and carbon isotope characteristics of Nujiang River water: Implications for CO₂ budgets of rock weathering in the Tibetan Plateau." *Science China Earth Sciences* 66(12): 2953-2970.*

Dear Editor,

I have read through the manuscript entitled “Refined weathering CO₂ budget of the Tibetan Plateau strongly modulated by sulphide oxidation” by Liu et al. This paper reported refined sulfide oxidation weathering flux with carbonate and silicate rocks. The biggest challenge for this theme in my opinion is that previous studies used model calculation by setting the majority of SO₄ sourced from sulfides, and therefore unable to accurately isolate the pyrite weathering flux, due to unavailable S isotopic data for many river basins (e.g., Torres et al., 2017, PNAS). Liu et al. used C-S isotopes for the first time accurately constrained the sulfide oxidation flux and CO₂ consumption in the entire Tibetan Plateau (TP), and finally found that the TP is an important carbon sink. This is a valuable contribution for understanding the carbon source and sink in the entire TP, as individual river basin in this region showed either a net source or sink. Their findings are particularly useful to assess the carbon cycle processes and fluxes in the tectonically active TP, a hot point region that has long been linked to tectonic uplift and the Cenozoic cooling.

The manuscript is generally well-written. The data support the conclusions. I have made some comments for the data interpretation that I think the authors can be solved at current version. Overall, I recommend the publication of this manuscript after minor modifications.

Comments:

Line 36: Change the “glacial” to “glaciers”

Line 70-71: “Both the river hydro-chemistry and Sr geochemistry suggest a highlighted carbonate dominated weathering regime on the plateau.” Why the Sr geochemistry suggest a carbonate dominated weathering? Are there metamorphic carbonate? The average $^{87}\text{Sr}/^{88}\text{Sr}$ for the TP is 0.71245, which is much higher than the limestone 0.709 or global evaporite 0.710-0.711. I agree with the hydro-chemistry interpretation that showing Ca-Mg-HCO₃- dominated ionic compositions, but I can not follow the $^{87}\text{Sr}/^{88}\text{Sr}$ statement. I suggest the authors to clarify it here for Sr isotopic description.

Line 81-85: I suggest to move these two sentences “Riverine sulfate origins are calculated with SO_4^{2-} concentration and $\delta^{34}\text{S}_{\text{SO}_4}$ value...higher f_{pyrite} for the studied river system indicates significant sulfuric acid involvements in the plateau weathering.” to after the Line 103 or other sections with sulfide weathering discussions. The authors started the quantified calculation regarding carbonate and silicate weathering contribution from Line 86-103, which provided basic information for endmember input. At this basis, then continue to calculate the sulfide contribution, and discuss the importance of sulfuric acid involvement weathering by sulfide oxidation.

Line 91-92: This is a little bit unclear for general readers. You may clarify this as “...are more complicated than rivers in other global geomorphology unites, because the TP are characterized by almost highest uplift rates, widespread mid-latitude glaciers, contrasting climate (across from >2000 mm/yr precipitation in southern Himalayan to extreme drought in plateau inland regions) and vegetations, and complex lithologies,

particularly the meta-carbonate weathering.

Line 94-96: Yes, this is novel to assess the sulfide weathering by systematically adding new sulfide endmember, so that can better constrain their contributions in the whole Tibetan Plateau.

Line 107: Change to “the area-averaged total chemical weathering rate...”, so that here is clear not regarding physical weathering rate.

Line 113: Change “at” to “extending to 6.6%”.

Line 114: Adding “A large spatial comparison indicates that the plateau river basins...”

Line 114-125: I think this comparison is very interesting, with which demonstrating that although the highly-eroded upstream plateau produced large amounts of debris with fresh mineral surface, the silicate weathering rates were still much lower than downstream area, suggesting a central role of climate (precipitation, runoff, temperature) together to enhance the silicate weathering rate. This is an additional supplementary to explain the uplift hypothesis.

Line 133-136: I agree with the impact from physical erosion, but here I think it maybe just the sulfide oxidation driven by rapid erosion accelerates the carbonate weathering, and finally results in higher CWR. This also agree with your sulfide weathering observations, and then you continue to introduce the CO₂ consumption from Line 137.

Line 139: It's better to add references for the glacier activity and erosion and OWP, as this is the first time to mention the glacier effects on OWP in the main text. You may

cite Cao et al., 2023, 901, 165842STOTEN. Their paper reported extensive sulfide oxidation in a typical glacial catchment in the NE Tibetan Plateau.

Line 140-143: Yes, this is one of the key evidence that showing most areas are affected by sulfide weathering in the whole Tibetan Plateau region when looking at the $\delta^{13}\text{C}_{\text{DIC}}$ values.

Line 146-148: This sentence “Partitioning the riverine sulfate sources... as a whole” is similar from in the line 90-92 “However, the river ...as revealed above”. I suggest to merge these two places in somewhere.

Line 164-166: It's better to add some discussion after the calculations. The result is very interesting in my opinion. You may say after “...sulfuric acid weathering effect is ignored.” Like this: “Therefore, although sulfide oxidation involved weathering in a single basin in the Tibetan Plateau reported either net CO₂ source or sink (references), which is all reasonable, depending on their sulfide oxidation rates and might also be the propositions of carbonate and silicate rocks, it is still an important net CO₂ sink from the comprehensive results of the entire Tibetan Plateau. We further find that the sulfuric acid dissolution ...”

Line 167-168: Change “Our study...quantifies...” to “ Our study, based on measured and compiled $\delta^{34}\text{S}_{\text{SO}_4}$ and solute compositions, for the first time accurately quantifies...”

Line 172-176: I agree with this because glacial catchment generally has higher erosion rates and thus pyrite exposure and oxidation. It's better to add a reference saying that

glacial basins reported higher erosion and sulfide oxidation, so that can well support your statement here.

Line 190-196: I think the biggest contribution of this paper is that the authors are able to quantify the sulfide weathering flux by using C and S isotopes in the tectonically active Tibetan Plateau, which is a challenge for previous studies by using model calculation which can not distinguish sulfide oxidation and gypsum dissolution due to no available S isotopes for many river basins, and therefore existing large uncertainties (see discussion by Torres et al., 2017, 114, 8716-8721, PNAS, <https://doi.org/10.1073/pnas.1702953114>). This can be add somewhere to highlight your contributions.

Prof. Dr. Fei Zhang, IEECAS.